Proteomics-based screening of AKR1B1 as a therapeutic target and validation study for sepsis-associated acute kidney injury

Li Lei 1
Ling Zaiqin 2
Wang Xingsheng 3
Zhang Xinxin 4
Li Yun Yunlee2022@163.com 5
Gao Guangsheng shenggao163@163.com 6
1 Intensive Care Unit, Shandong Public Health Clinical Center, Shandong University , Jinan , China
2 Department of Tubercular Medicine, Shandong Public Health Clinical Center, Shandong University , Jinan , China
3 Department of Emergency, Beijing Chaoyang Hospital, Capital Medical University , Beijing , China
4 Department of Emergency Medicine, Fuyang People’s Hospital of Anhui Medical University , Fuyang , China
5 Intensive Care Unit, Central Hospital Affliated to Shandong First Medical University , Jinan , China
6 Neurological Intensive Care Unit, Central Hospital Affliated to Shandong First Medical University , Jinan , China
Dong Peixin
Electronic publication date: 2024 Jan 2
Publication date: 2024
Volume: 12
Electronic Location ID: e16709
Received 2023 Sep 20; Accepted 2023 Dec 1
Copyright: ©2024 Li et al.
Copyright year: 2024
Copyright holder: Li et al.
License: This is an open access article distributed under the terms of the Creative Commons Attribution License, which permits unrestricted use, distribution, reproduction and adaptation in any medium and for any purpose provided that it is properly attributed. For attribution, the original author(s), title, publication source (PeerJ) and either DOI or URL of the article must be cited.
License URL: https://creativecommons.org/licenses/by/4.0/

Keywords: Proteomics, AKR1B1, Sepsis, Acute kidney injury, Therapy

Funding: Science and Technology project of Jinan Mulnipal Health Commission 2023-1-07 This work was supported by the Science and Technology project of Jinan Mulnipal Health Commission (No. 2023-1-07). The funders had no role in study design, data collection and analysis, decision to publish, or preparation of the manuscript.

==============================
Background

Sepsis and sepsis-associated acute kidney injury (SA-AKI) pose significant global health challenges, necessitating the development of innovative therapeutic strategies. Dysregulated protein expression has been implicated in the initiation and progression of sepsis and SA-AKI. Identifying potential protein targets and modulating their expression is crucial for exploring alternative therapies.

Method

We established an SA-AKI rat model using cecum ligation perforation (CLP) and employed differential proteomic techniques to identify protein expression variations in kidney tissues. Aldose reductase (AKR1B1) emerged as a promising target. The SA-AKI rat model received treatment with the aldose reductase inhibitor (ARI), epalrestat. Blood urea nitrogen (BUN) and creatinine (CRE) levels, as well as IL-1β, IL-6 and TNF-α levels in the serum and kidney tissues, were monitored. Hematoxylin-eosin (H-E) staining and a pathological damage scoring scale assessed renal tissue damage, while protein blotting determined PKC (protein kinase C)/NF-κB pathway protein expression.

Result

Differential proteomics revealed significant downregulation of seven proteins and upregulation of 17 proteins in the SA-AKI rat model renal tissues. AKR1B1 protein expression was notably elevated, confirmed by Western blot. ARI prophylactic administration and ARI treatment groups exhibited reduced renal injury, low BUN and CRE levels and decreased IL-1β, IL-6 and TNF-α levels compared to the CLP group. These changes were statistically significant (P < 0.05). AKR1B1, PKC-α, and NF-κB protein expression levels were also lowered in the ARI prophylactic administration and ARI treatment groups compared to the CLP group (P < 0.05).

Conclusions

Epalrestat appeared to inhibit the PKC/NF-κB inflammatory pathway by inhibiting AKR1B1, resulting in reduced inflammatory cytokine levels in renal tissues and blood. This mitigated renal tissue injuries and improved the systemic inflammatory response in the severe sepsis rat model. Consequently, AKR1B1 holds promise as a target for treating sepsis-associated acute kidney injuries.

Introduction

Sepsis, characterised by potentially devastating organ failure, stems from an aberrant host response to infections (Singer et al., 2016), impacting millions globally each year. Sepsis-associated acute kidney injury (SA-AKI) represents a potentially fatal consequence of septic shock or sepsis specifically affecting the kidneys. It results in a gradual loss of renal function, meeting the criteria established by the Global Kidney Disease Prognosis Organization (KDIGO) for AKI while excluding other potential causes of kidney damage (Bellomo et al., 2017). Studies indicate that SA-AKI constitutes 45%–70% of all AKI cases (Uchino et al., 2005). Furthermore, SA-AKI tends to be more severe and exhibits a higher fatality rate compared to other AKI types (Bagshaw, George & Bellomo, 2008). The complete understanding of the pathophysiological mechanisms of SA-AKI remains elusive, with existing theories largely reliant on autopsy data and animal models. Microcirculatory abnormalities, inflammation and metabolic reprogramming are widely presumed as the three primary mechanisms contributing to SA-AKI development (Peerapornratana et al., 2019). Several studies have demonstrated that antibiotics, vasopressors, fluid resuscitation, pharmacological inhibitors of signalling pathways, renal replacement therapy and phytochemicals have shown potential in treating sepsis and SA-AKI. However, there are currently no established, effective or specific techniques for preventing or treating SA-AKI. Dexamethasone, for instance, demonstrated a reduced need for kidney replacement therapy in patients with sepsis (Jacob et al., 2015). A phase II trial indicated long-term kidney benefits and lower mortality with the anti-inflammatory recombinant alkaline phosphatase (Pickkers et al., 2018). Regarding haemodynamic and oxygen delivery, studies using angiotensin 2 (Tumlin et al., 2018) and levosimendan (Tholén, Ricksten & Lannemyr, 2021) suggest potential renal protection, yet effective and specific strategies for preventing or treating SA-AKI remain elusive. Thus, there is a critical need to explore novel medications and therapeutic targets for SA-AKI.

Differential proteomics, a pivotal aspect of proteomics research, focuses on identifying factors leading to proteomic variances between samples, thereby elucidating and validating proteomic changes in physiological and pathological processes. This technique aids in identifying differentially expressed proteins pivotal to pathogenesis, serving as a tool to determine disease biomarkers and potential therapeutic targets (Song et al., 2021; Su et al., 2014).

The application of proteomics to human and animal AKI models has unveiled numerous genes and proteins emerging as biomarkers and therapeutic targets (Devarajan, 2008; Thongboonkerd, 2004; Thongboonkerd, 2005).

In this study, a label-free LC-MS/MS proteomics technique was employed to scrutinise differential proteins in renal tissues of the rat SA-AKI model and the sham-operated group. Subsequently, raw letter analysis and a literature review identified aldose reductase (AKR1B1) as a potential therapeutic target for subsequent animal experiments.

Aldose reductase (AKR1B1), the first rate-limiting enzyme involved in the polyol pathway, is linked to the pathogenesis of diabetes-related issues like cataracts, neuropathy, retinopathy and nephropathy (Kato et al., 2009). Given its involvement in inflammatory diseases, such as sepsis, AKR1B1 has garnered significant research interest in recent years (Rakowitz et al., 2007). A prior study demonstrated that AKR1B1 inhibition reduced inflammatory responses induced by cecum ligation puncture in mice (Reddy, Srivastava & Ramana, 2009). Additionally, Takahashi et al. (2012) observed that the AKR1B1 inhibitor, fidarestat, improved LPS-induced acute renal damage and reduced mortality. These findings underscore AKR1B1 as a potential therapeutic target for SA-AKI.

In this study, AKR1B1 inhibition effectively mitigated SA-AKI symptoms, providing valuable insights into potential therapeutic targets for SA-AKI.

Material and Methods

Experimental Procedures

Experimental animals

Specific pathogen-free (SPF)-grade male Sprague-Dawley (SD) rats (220–260 g, 6 weeks old), obtained from Speifu Biotechnology Co., Ltd (Beijing, China) with Animal Certificate of Conformity No. SCXK (Beijing, China) 2019-0010, were acclimated for 1 week at the Central Hospital Affiliated to Shandong First Medical University SPF-level experimental animal centre. The conditions included an ambient humidity of 50 ± 5%, ambient temperature of 23 ± 1 °C and 12 h light/dark cycle. Rats were provided irradiated feed and sterile water. After the experiment, SD rats were euthanised via intraperitoneal pentobarbital sodium (30 mg/kg) injection. All animal handling methods adhered to ethical standards and received approval from the Laboratory Animal Welfare and Ethics Committee at the Central Hospital Affiliated with Shandong First Medical University (JNCH 2022-8).

Modelling

Moderate and severe sepsis rat models were constructed using the cecum ligation perforation (CLP) method (Rittirsch et al., 2009). Rats were anaesthetised with 1% sodium pentobarbital (40 mg/kg). The lower and middle abdomen were longitudinally incised along the median axis to easily access the abdominal cavity. The cecum was divided and excised before being ligated with 4-0 sutures from the end of the cecum to one-half (moderate sepsis) or three-quarters (severe sepsis) of the ileocecal valve, respectively. Two punctures were made in the ligated distal cecum with an 18G needle. The treated appendix was retracted into the abdominal cavity. Subsequently, sutures were placed and the surgical region was disinfected again using iodophor. Subcutaneous injections of pre-warmed saline (3 ml/100 g, 37 °C) were used for resuscitation. The sham-operated group (sham group) did not undergo cecum ligation and perforation, and their cecum was bluntly removed and retracted into the abdominal cavity.

Grouping

Differential proteomics and validation experiments

A total of 18 rats were randomly divided into three groups, each containing six rats: group S (sham-operated group, Sham), group A (moderate sepsis group, CLP1/2) and group B (severe sepsis group, CLP3/4). Rat models were developed for 24 h, and kidney tissues were collected for proteomics and validation experiments.

AKR1B1 therapeutic target study

A total of 32 rats were randomly categorised into four groups, each containing eight animals: Sham, CLP, CLP+pre-ARI (administered medicine before modelling) and CLP+post-ARI (administered medicine after modelling) groups. Serum and renal tissue samples were collected 24 h post modelling. The CLP+pre-ARI group received epalrestat (Yangtze River Pharmaceutical Group, China), dissolved in 0.5% sodium carboxymethyl cellulose (v/v; Na-CMC), at a concentration of 100 mg/kg/d, for one week before CLP, while the CLP+post-ARI group received epalrestat 2 h after CLP via gavage. The Sham and CLP groups received equivalent volumes of 0.5% Na-CMC solution via gavage, once a day for a week before CLP. The epalrestat dose was determined based on previous studies (Gao et al., 2019; Li et al., 2016; Yang et al., 2019).

Differential Proteomics

Protein treatment

Kidney tissue samples were homogenised, and proteins were extracted. Total protein concentration was determined using the BCA technique. Equal protein amounts were enzymatically digested, and peptides were desalted and eluted with 80% acetonitrile (ACN). Peptide quantification utilised the BCA kit (Beyotime Biotech Inc, Shanghai, China).

Liquid chromatography-mass spectrometry (LC-MS)

Liquid chromatography utilised mobile phase A (0.1% formic acid + 2% ACN dissolved in water) to dissolve the peptides, which were then separated using a NanoElute ultra-high performance liquid phase device (Bruker timsTOF Pro; Bruker, Bremen, Germany). Mobile phase B constituted 0.1% formic acid + 100% ACN. Gradient parameters for liquid chromatography were 0–70 mins: 6%–24% B; 70–84 mins: 24%–32% B; 84–87 mins: 32%–80% B; and 87–90 mins: 80% B. The UHPLC technology was used to isolate the peptides, and the samples were injected into the capillary ion source for ionisation and assessed using timsTOF Pro mass spectrometry technique. An ion source voltage of 1.6 kV and high-resolution TOF identified and characterised peptides. The secondary mass spectrometry scanning range was fixed at 100-1700. For data acquisition, the parallel accumulated serial fragmentation (PASEF) mode was employed. Following the collection of primary mass spectrum data, ten PASEF mode secondary spectral acquisitions with parent ion charge values, ranging from zero to 5, were performed. To minimise repetitive scanning of the parent ions, dynamic exclusion time was fixed at 30 s.

Database search

Maxquant was used for database searches (v1.6.15.0). The theoretical and mass spectroscopy-obtained secondary spectrum maps were compared. The identified protein-specific peptides were used to retrieve the protein-related information using the following search parameters: Rattus norvegicus 10116 PR 20201214.fasta (29,940 sequences) served as the database, while a different inverse library was included for computing the false positive rate (FPR) owing to random matches. A second common contamination library was also included in the database to decrease the potential effects of the contaminating proteins on the results of the identification process. Trypsin/P digestion was employed, where the number of missed cut sites was fixed at two, while the minimal peptide length was fixed at seven amino acid residues and the maximal number of peptide modifications was fixed at five. Moreover, the primary parent ion’s mass error tolerance was set at 20 ppm, while the mass error tolerance of the secondary fragment ion was fixed to 20 ppm. A fixed modification was established as the carbamidomethyl (C) alkylation of cysteine, while the variable modifications, such as methionine oxidation and acetylation of protein N-terminals, were included. False discovery rates (FDR) for PSM and protein identifications were set at 1%.

Differential protein screening

The t-test assessed, relative quantitative values, with a P value ≤0.05 indicating significance. Differential expression >1.5 indicated significant upregulation, while values <1/1.5 indicated significant downregulation (Chen et al., 2022; Melenovsky et al., 2018). The screened proteins were further validated using Western blotting.

AKR1B1 Therapeutic Target Study

Histopathological examination

Standard procedures were employed for kidney tissue processing. Paraffinised kidney sections were collected, dehydrated, de-paraffinised and stained using HE staining. The light microscopic images were analysed and collected for data analysis.

The renal tubular injury was scored by two pathologists in a double-blind manner using criteria from a previous study (Pieters et al., 2019). Scoring included 0 for no injury; 1 for ≤25%; 2 for 26%–50%; 3 for 51%–75%; and 4 for >75%. Five differing viewing fields were selected for scoring each pathological section, and data were statistically analysed.

Blood serum urea nitrogen (BUN) and creatinine (CRE) assay

Blood serum urea nitrogen levels and creatinine levels were determined using the Urea Nitrogen (BUN) test kit (urease method) (Nanjing Jiancheng Bioengineering Institute, Nanjing, China) and creatinine (CRE) assay kit (sarcosine oxidase method) (Creatinine (Cr)), respectively. Absorbance values were measured at OD values of A640 and A546 using the enzyme standardisation instrument, following the manufacturer’s instructions. Finally, the serum BUN and CRE levels were calculated using the recommended formulae.

Enzyme-linked Immunosorbent Assay (ELISA)

Inflammatory cytokine levels, such as TNF-α (EK382; Shanghai Lianke Biological Co., LTD), IL-1β (EK301B; Shanghai Lianke Biological Co., Ltd., Shanghai, China) and IL-6 (EK306, Shanghai Lianke Biological Co., Ltd., Shanghai, China), in rat serum and kidney tissues were determined using commercial ELISA kits (Boster Bioengineering Co., Ltd., Wuhan, China). An enzyme standardisation device was employed to determine the absorbance (OD) values of the samples at 450 nm, and the ELISA Calc software was used for standard curve plotting and concentration calculation.

Western blot

Protein samples from homogenised kidney tissue samples were subjected to the BCA technique to determine total protein concentrations. Then, these protein samples were electrophoresed using sodium dodecyl sulfate-polyacrylamide gel electrophoresis (SDS-PAGE) before transferring onto PVDF membranes. Subsequently, the membranes were placed in a container with a 5% BSA blocking solution and shaken for 2 h at room temperature. The PVDF membrane was then buffer rinsed and treated with primary antibodies (AKR1B1 1:500; β-actin 1:1000; PKC-α 1:200; NF-κB p65 1:200; SANTA CRUZ) and incubated overnight at 4 °C. Following this, the PVDF membrane was buffer rinsed and incubated in the presence of secondary antibodies (m-IgGκ BP-HRP 1:5000; SANTA CRUZ) for 60 mins at room temperature in the shaker. The ECL developer was then used to develop the membranes, and Image J software evaluated protein sample bands.

Real-time quantitative PCR (RT-qPCR)

AKR1B1 gene expression in rat renal tissues from Sham, CLP, CLP+pre-ARI and CLP+post-ARI was determined using RT-qPCR. Total RNA was extracted from 50 mg of each tissue sample stored at −80 °C, following the RNAex Pro RNA Extraction Reagent (#AG21102; Accurate Biotechnology, Shenzhen, China) protocol. The quality of the RNA was confirmed using a NanoDrop One spectrophotometer (Thermo Fisher Scientific, Waltham, MA, USA) with an A260/280 ratio within the range of 1.8−2.0. Genomic DNA digestion and reverse transcription were performed using the Evo M-MLV RT Mix Kit and the gDNA Clean for qPCR kit (#AG77128; Accurate Biotechnology, Shenzhen, China). Subsequently, 1.0 µg of RNA was converted into cDNA using oligo(dT) primers in a 20 µL reaction mixture. Reverse transcription was performed at 37 °C for 15 mins and 85 °C for 5 s. cDNA was stored at −80 °C. Then, qPCR was conducted using a LightCycler 96 Real-Time PCR Detection System with a 20 µL reaction mixture constituting 2 µL of cDNA, 10 µL of 2X SYBR Green Pro Taq HS Premix (#AG11701; Accurate Biotechnology, Shenzhen, China) and gene-specific primers (AXYGEN, PCR-0208-C). The PCR programme included an initial denaturation step at 95 °C for 30 s, followed by 40 cycles at 95 °C for 5 s and 60 °C for 30 s. Melting curve analysis was performed at 95 °C for 15 s, 60 °C for 1 min and 95 °C for 5 s. The dissolution curve was unimodal. There was no Cq value for the amplification of the No Template Control (NTC). Gene-specific primers (Table 1) were designed with amplicon lengths of approximately 100 bp, confirmed through NCBI blasting. Fold changes in RNA abundance were calculated using the 2 ˆ(−ΔΔCT) method, with β-actin serving as the internal reference and slope fluctuate between −3.59 and 3.1, R2 ≥0.9. A limit of detection was performed, with LOD = 2.5 targeting molecules with a 95% confidence interval. Data analysis excluded any technical replicates with significant deviations from the other two values in three technical replicates. Each group had eight biological replicates, and each sample underwent three technical replicates to ensure robust results.

Table 1 Gene-specific primers for real-time RT-PCR.

Gene	Forward primer	Reverse primer	
AKR1B1	5′-CTCAACAACGGCACCAAGATG-3′	5′-CCATGTCGATAGCAACCTTCAC-3′	
β-Actin	5′-CACCCGCGAGTACAACCTTC-3′	5′-CCCATACCCACCATCACACC-3′	

Statistical Analysis

Data analysis and visualisation were conducted using SPSS 25.0 (SPSS Inc., Chicago, IL, USA) and GraphPad Prism 9.0 (GraphPad, La Jolla, CA, USA) software. Non-normally distributed data were analysed using the Kruskal–Wallis test, while normally distributed quantitative data were presented as mean ± SD. Additionally, one-way analysis of variance (ANOVA) was used for comparing data from different samples with normal distributions, and statistical significance was considered at P < 0.05.

Results

Proteomics differential protein screening

A total of 5,367 proteins were detected through proteomics, with 4448 proteins quantified. Differential expression analysis (>1.5-fold or <1/1.5-fold) was conducted on kidney samples from septic and sham-operated group(s). Comparative analysis revealed distinctive protein expression patterns in moderate sepsis (A) and severe sepsis (B) rat AKI models, as illustrated in Fig. 1.

Figure 1 Differential protein screening.

(A) Differential protein expression (B) Differential protein volcano maps (C) Differential protein heat maps. S, sham group, A: moderate sepsis; B: severe sepsis.

The results indicated an upregulation of 17 proteins and a significant downregulation of seven proteins in both moderate and severe sepsis groups compared to the sham group. An extensive literature review was performed for each identified protein, considering unique peptide numbers and differential protein multiples. Proteins relevant to the research focus were selected for further study (Table 2).

Table 2 In both the moderate sepsis (A) and severe sepsis (B) rat S-AKI models, 17 proteins were upregulated and seven were downregulated.

Protein description	Gene name	Regulated Type	
Parvalbumin alpha	Pvalb	Down	
Apolipoprotein A-IV	Apoa4	Down	
Metallothionein-1	Mt1	Down	
Apolipoprotein A-II	Apoa2	Down	
Heme oxygenase 1	Hmox1	Down	
Polypeptide N-acetylgalactosaminyltransferase	Galnt3	Down	
Dimethylglycine dehydrogenase, mitochondrial	Dmgdh	Down	
Endothelial cell-selective adhesion molecule	Esam	Up	
Adipocyte-type fatty acid-binding protein	Fabp4	Up	
Serine protease inhibitor A3N	Serpina3n	Up	
Carboxylic ester hydrolase	LOC501233	Up	
Alpha-1-acid glycoprotein	Orm1	Up	
Mitochondrial ribonuclease P catalytic subunit	Prorp	Up	
Periplakin OS=Rattus norvegicus	Ppl	Up	
L-serine ammonia-lyase	Sds	Up	
Serine protease inhibitor A3M	Serpina3m	Up	
Aldo-keto reductase family 1, member B8	Akr1b8	Up	
Protein S100-A4	S100a4	Up	
Alpha-2-macroglobulin	A2m	Up	
Aldo-keto reductase family 1 member B1	Akr1b1	Up	
Neutrophil gelatinase-associated lipocalin	Lcn2	Up	
Chloride channel protein	Clcnka	Up	
Lipopolysaccharide-binding protein	Lbp	Up	
Macrophage-capping protein	Capg	Up	

AKR1B1 is highly expressed in the renal tissue of the SA-AKI rat model

To identify potential therapeutic targets for SA-AKI, 24 proteins were further screened. AKR1B1 emerged as a promising candidate due to its statistically significant expression difference between the B/S groups (P = 6.56272939732895E−06) and its documented association with inflammation (Chen et al., 2018a; Miláčková et al., 2017; Wang et al., 2023). As studies reporting on AKR1B1 and its involvement in sepsis are scarce, AKR1B1 was selected as the candidate molecule. Western blot analysis confirmed an elevated AKR1B1 protein level in CLP1/2 and CLP3/4 compared to the sham group, aligning with proteomic findings (Fig. 2).

Figure 2 High expression of AKR1B1 in the renal tissue of the SA-AKI rat model.

(A) Western blot analysis of AKR1B1 expression level in renal tissue from the Sham, CLP1/2 and CLP3/4 groups. (B) Quantitative analysis of AKR1B1. *** P < 0.001 vs Sham, **** P < 0.0001 vs Sham. Sham: sham-operated group; CLP1/2: moderate sepsis group; CLP3/4: severe sepsis group.

Epalrestat reduces renal function impairment in rats

In evaluating the protective effect of epalrestat, a CLP-induced SA-AKI rat model was established. The results showed a significant increase in serum BUN and CRE levels in the CLP group compared to the sham group at 24 h post-operation (P < 0.05) (Figs. 3A and 3B). Both CLP+pre-ARI and CLP+post-ARI groups exhibited a decrease in BUN and CRE levels, with statistically significant differences compared to the CLP group (P < 0.05) (Figs. 3A and 3B).

Figure 3 Changes in renal function parameters in the serum from the sham, CLP, CLP+pre-ARI and CLP+post-ARI groups.

Serum BUN (A) and creatinine (B) levels were measured to evaluate renal function. # P < 0.05 vs Sham, * P < 0.05 vs CLP, ** P < 0.01 vs CLP. mean ± SD, one-way ANOVA, double-tailed unpaired t-test, n = 8/group.

Epalrestat decreases inflammatory cytokine levels in the serum and renal tissues in the severe sepsis rat model

Examination of inflammatory cytokine levels revealed a significant elevation of TNF-α, IL-1β and IL-6 in renal tissues and serum homogenates of the CLP group compared to the sham group (P < 0.05). Epalrestat administration, both before and after CLP, partially reversed IL-6, IL-1β and TNF-α expressions, with statistically significant differences (P < 0.05, Figs. 4A and 4B).

Figure 4 Changes in the inflammatory cytokine levels in the serum and renal tissues.

(A) Serum levels of TNF-α, IL-1β and IL-6. (B) Renal tissue levels of TNF-α, IL-1β and IL-6. # P < 0.05 vs Sham, * P < 0.05 vs CLP, ** P < 0.01 vs CLP. mean ± SD, one-way ANOVA, double-tailed unpaired t-test, n = 8/group.

Epalrestat may protect against SA-AKI

Histopathological examination at 24 h post-operation demonstrated significant renal tissue injuries in the CLP group, including oedema, vacuolar degeneration, tubular epithelial swelling, tubular necrosis and brush border loss, compared to the sham group. CLP+pre-ARI and CLP+post-ARI groups exhibited varying degrees of reduction in renal tissue injury, with a statistically significant decrease in renal histopathology scores compared to the CLP group (P < 0.05) (Figs. 5A and 5B). These findings suggested that epalrestat could contribute to protecting against AKI in severe sepsis.

Figure 5 The degree of renal tubular injury was evaluated using HE staining.

(A) HE staining. 400 ×. Scale bar = 100µM. (B) Kidney histopathological score. # P < 0.05 vs Sham, * P < 0.05 vs CLP, ** P < 0.01 vs CLP.

Epalrestat inhibits the activation of the AKR1B1/PKC/NF-κB pathway

Analysis of AKR1B1 mRNA levels revealed a significant increase in the CLP group (P < 0.05), while the ARI prophylactic administration and ARI treatment groups showed a significant decrease compared to the CLP group (P < 0.05) (Fig. 6A). Western blot analysis further indicated elevated levels of AKR1B1, PKC-α and NF-κB p65 proteins in the CLP group compared to the sham group (P < 0.05). In contrast, the CLP+pre-ARI and CLP+post-ARI groups exhibited reduced levels of these proteins (P < 0.05), with statistically significant differences compared to the CLP group (P < 0.05) (Figs. 6B and 6C).

Figure 6 Epalrestat inhibits the activation of the AKR1B1/PKC/NFκB pathway.

(A) qRT-PCR analysis showing the relative mRNA level of AKR1B1 in renal tissue from the Sham, CLP, CLP+pre-ARI and CLP+post-ARI groups. (B) Western blot analysis showing the relative protein expression level of AKR1B1, PKC-α and NFκB p65. # P < 0.05 vs Sham, * P < 0.05 vs CLP, ** P < 0.01 vs CLP, **** P < 0.0001 vs CLP. mean ± SD, one-way ANOVA, double-tailed unpaired t-test, n = 8/group.

Discussion

Proteomics, exploring the diverse proteomes, post-translational modifications, protein-protein interactions and biological functions, is integral in unravelling the intricate mechanism underlying diseases. Comparative proteomic analyses between healthy and pathological states serve as valuable tools in identifying unique protein molecules, potentially acting as molecular markers for early disease detection and as future drug targets (Ho, Dart & Rigatto, 2014). The application of proteomics, particularly mass spectrometry techniques, to human and animal models of AKI has illuminated the pathophysiology of AKI and revealed novel genes and proteins as biomarkers and therapeutic targets, including the Bcl-2 family proteins and kidney injury molecule 1 (Kim-1) (Hoffmann et al., 2002; Ichimura et al., 1998). Current advances in multi-omics technologies have enabled a comprehensive analysis of SA-AKI, contributing to the prevention, diagnosis, staging and treatment of SA-AKI (Qiao & Cui, 2022).

In this study, proteomic analysis identified 24 differential proteins associated with SA-AKI. Notably, proteins like serine protease inhibitor (Serine protease inhibitor) and heme oxygenase-1 (Hmox1), previously explored in sepsis, were among the identified targets (Chen et al., 2018b; Li et al., 2018; Ryter, 2021; Shutong et al., 2022; Yang et al., 2020). On the other hand, aldose reductase (AR, AKR1B1) emerged as a novel candidate for SA-AKI treatment, showcasing the potential of proteomics in uncovering unexplored therapeutic targets.

AKR1B1, an NADPH-dependent oxidoreductase from the aldo-keto reductase superfamily (AKR), holds significance in the polyol pathways of gluconeogenesis (Sonowal & Ramana, 2021). In the gluconeogenesis pathways, it uses NADPH as a cofactor to convert glucose to sorbitol, inducing chronic diabetic complications in hyperglycemic states (Quattrini & La Motta, 2019). Traditionally, studies in the context of the aetiology of diabetic complications, AKR1B1 have been less explored in SA-AKI (Thakur et al., 2021). Additionally, studies have reported that aldose reductase inhibitors (ARI) could be used to prevent and treat diabetic retinopathy, neuropathy and nephropathy (Chang, Shieh & Petrash, 2019; He et al., 2019; Schemmel, Padiyara & D’Souza, 2010; Sekiguchi et al., 2019).

Recently, some studies explored the role of AKR1B1 in mediating various inflammatory complications. As a result, there is an increase in the number of studies investigating the development and application of ARI in the treatment of inflammatory complications, diabetes and cancers. ARI has also been shown to prevent asthma, sepsis, uveitis and colon and breast cancer (Pandey, 2015; Shukla et al., 2017; Wu et al., 2017; Yadav et al., 2011). Notably, several studies have utilised ARI for the treatment of COVID-19 infection (Gaztanaga et al., 2021).

It has been shown that AKR1B1 effectively mediates inflammatory signalling by decreasing the levels of reactive oxygen species (ROS)-induced lipid peroxidation-derived lipid aldehydes like 4-hydroxy-trans-2-nonenal (HNE) and its glutathione coupling (e.g., GS-HNE) to the respective alcohols (Srivastava et al., 1998; Srivastava et al., 2000). AKR1B1 inhibition has been reported to effectively inhibit the inflammatory signalling reactions induced by various stimuli, such as endotoxins, cytokines, growth factors, allergens, high glucose and autoimmune responses. Moreover, AKR1B1 inhibition also prevents the oxidative stress-induced activation of NF-κB and activator protein-1 (AP-1) (Ramana, Bhatnagar & Srivastava, 2004a; Ramana et al., 2004b), which play important roles in the expression of chemokines, inflammatory factors and other inflammatory markers (Bhattacharyya, Biswas & Datta, 2004). Ramana & Srivastava (2006) demonstrated that ARI inhibited the LPS-induced activation of TNF-α, IL-1β, IL-6, monocyte chemoattractant-1 (MCP-1) in mouse peritoneal macrophages protein-1 and other inflammatory cytokines, suggesting a significant activation of PKC in peripheral macrophages by LPS but ARI prevented this LPS-induced PKC activation. Garg et al. (2012) noted that PKC was involved in activating the NF-κB inflammatory pathway and the secretion of many pro-inflammatory cytokines. ARI inhibits PKC activity by reducing the PLC and PKC phosphorylation levels, which, in turn, leads to the downstream inactivation of the IKK/IκB/NF-κB inflammatory pathway (Zeng et al., 2013).

Acute kidney injury (AKI), a severe complication of sepsis, is characterised by high mortality and poor prognosis. Despite significant advances in treatment, monitoring and medical support technologies, the pathophysiological mechanisms of SA-AKI remain unclear. For decades, SA-AKI was speculated to be driven by intrarenal hypoxia and ischaemic injury, owing to inadequate renal perfusion; however, this hypothesis is yet to be experimentally validated. On the other hand, experimental evidence suggested that renal blood flow was preserved or even increased. In the past few years, the immune and inflammatory mechanisms have garnered increasing scientific attention (Langenberg et al., 2014; Maiden et al., 2016). Animal experimental findings were conducted to identify different drugs that can inhibit oxidative stress, inflammation and apoptosis, and which can be used for treating SA-AKI (Hu et al., 2022; Salari et al., 2022; Xie et al., 2022).

Here, AKR1B1 was recognised as a therapeutic target for SA-AKI using a differential proteomics approach. Epalrestat, a commercial ARI, was employed as an intervention in a rat sepsis model, demonstrating significant reductions in inflammatory factors TNF-α, IL-1β and IL-6. Furthermore, compared to the CLP group, AKI was significantly decreased to varying degrees in both the ARI intervention groups. However, Takahashi et al. (2012) reported that ARI did not affect the AKR1B1 expression levels despite its ameliorative effect on LPS-induced systemic inflammatory response syndrome in AKI. In this study, epalrestat was observed to inhibit the mRNA and protein expression levels of AKR1B1, which is an upstream regulator of PKC activation. PKC inhibition also inhibits the NF-κB pathway, which suppresses inflammation and improves SA-AKI. However, as previously stated, many complex signalling pathway interactions exist between AKR1B1 and PKC, and between PKC and NF-κB, which could not be confirmed in this study. Hence, in-depth studies exploring the detailed mechanisms of AKR1B1 as a therapeutic target for SA-AKI need to be performed.

Conclusions

This study employed a proteomic approach to unveil the significant upregulation of AKR1B1 expression in SA-AKI. Subsequent animal experiments confirmed that inhibiting AKR1B1 effectively suppressed the activation of the PKC/NF-κB inflammatory pathway, resulting in decreased expression levels of inflammatory cytokines in both renal tissue and serum samples. Thus, the damage to the kidney tissue in severe sepsis model rats was alleviated. The primary objective of this research was to attenuate inflammatory cytokine expression levels in the serum and kidney tissues in the severe sepsis rat model. The identification of AKR1B1 as a different therapeutic target not only sheds light on the pathogenesis of SA-AKI but also introduces a novel avenue for clinical prevention in sepsis-associated complications. Epalrestat, a commercial ARI, emerged as a promising candidate for treating SA-AKI. However, its effectiveness needs to be further investigated owing to the complexity of the pathogenesis and pathophysiology of sepsis and SA-AKI.

Supplemental Information

Data S1 Raw data

Click here for additional data file.

Supplemental Information 2 ARRIVE Checklist

Click here for additional data file.

Supplemental Information 3 MIQE Checklist

Click here for additional data file.

Additional Information and Declarations

Competing Interests

Author Contributions

Animal Ethics

Data Availability

The authors declare there are no competing interests.

Lei Li conceived and designed the experiments, performed the experiments, prepared figures and/or tables, authored or reviewed drafts of the article, and approved the final draft.

Zaiqin Ling analyzed the data, prepared figures and/or tables, and approved the final draft.

Xingsheng Wang performed the experiments, authored or reviewed drafts of the article, and approved the final draft.

Xinxin Zhang performed the experiments, prepared figures and/or tables, and approved the final draft.

Yun Li performed the experiments, authored or reviewed drafts of the article, and approved the final draft.

Guangsheng Gao conceived and designed the experiments, authored or reviewed drafts of the article, and approved the final draft.

The following information was supplied relating to ethical approvals (i.e., approving body and any reference numbers):

This study approved by Laboratory animal welfare and ethics committee in Central Hospital Affiliated to the Shandong First Medical University.

The following information was supplied regarding data availability:

The raw measurements are available in the Supplementary File.

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
