# Peer review of "Proteomics-based screening of AKR1B1 as a therapeutic target and validation study for sepsis-associated acute kidney injury"

_PeerJ, doi:10.7717/peerj.16709_

## Round 0.1 · original submission · Major Revisions

Issues that need to be revised or clarified:

Major issues:
1. The conclusions of the Abstract (lines 52-55), as well as the Conclusions section (lines 397-401) should be modified, because their results indicated the possibility that epalrestat suppressed the activation of PKC/NFκB inflammatory pathway by inhibiting AKR1B1, more detailed research was required to validate this possibility. Therefore, the tone of these parts needs to be softened, with more emphasis on the possibilities.
2. Many of the results are described in insufficient detail. For example, lines 280-281: [It showed a significant upregulation and its expression level was verified using a Western blot]: the expression of AKR1B1 was upregulated in which group? Please clarify this.

Minor issues:
1. In this paper, [NFκB] should be revised to [NF-κB].
2. Line 50: [AKR1B1, PKC-α, and NFκB expression levels were]: Add [protein] before [expression].
3. Line 61: [, that affects] or [which affects]? The language should be improved.
4. Line 131: The meaning of [pre-ARI, or post-ARI groups] should be explained.
5. Line 195: [lung sections]? This appears to be a spelling error.
6. Line 274: [upregulated expression of 17 proteins, while a significant downregulation was noted in 9 proteins] means that these 17 proteins increased in the SEVERE sepsis group, while 9 proteins decreased in SEVERE sepsis, relative to the MODERATE sepsis group? Please check if these are correct.
7. Figure 2: CLP1/2 or CLP3/4 means what? In the Fig. 2 legend, only CLP, CLP+pre-ARI and CLP+post-ARI were shown. Please address this inconsistency.
8. Line 289 and 298: A summary should be added here.
9. Line 176: [relatedsecondary]?
10. Line 243: [for determining] should be [to determine].
11. Line 265: [The reverse transcription processed] should be [The reverse transcription was processed].
12. Line 266 [The cDNA store] should be [The cDNA was stored].
13. Line 382 [which investigated] should be [that investigated].
14. Line 390: [rections]? It should be [reactions].

**Language Note:** The review process has identified that the English language must be improved. PeerJ can provide language editing services - please contact us at [email protected] for pricing (be sure to provide your manuscript number and title). Alternatively, you should make your own arrangements to improve the language quality and provide details in your response letter. – PeerJ Staff

Reviewer 1 ·

Basic reporting

This manuscript had clear and unambiguous, professional English used throughout. Relevant prior literature were appropriately referenced. The structure of the article conform to an acceptable format. The results were relevant to the hypothesis.

Experimental design

Original primary research was within Aims and Scope of the journal. The submission clearly define the research question.

Validity of the findings

No comment

Additional comments

This is a well written paper that presents interesting findings, the statistical analysis was well done and appropriate.

Reviewer 2 ·

Basic reporting

In this study, researchers aimed to develop a rat model of sepsis-associated acute kidney injury (SA-AKI) and identify potential protein targets for therapeutic interventions. They used the cecum ligation perforation (CLP) method to induce SA-AKI in the rats and employed differential proteomic techniques to identify differentially expressed proteins in the kidney tissues of the model rats. The results showed significant changes in protein expression in the renal tissues of the SA-AKI rat model, including upregulation of AKR1B1. Treatment with epalrestat resulted in decreased renal injury, as evidenced by H-E staining and pathological injury scores. The treatment also led to a decrease in BUN and CRE levels and a reduction in IL-1β, IL-6, and TNF-α levels in both the kidney tissues and blood. Further analysis revealed that AKR1B1 mRNA levels were significantly reduced in the epalrestat-treated groups compared to the CLP group, and the expression levels of AKR1B1, PKC-α, and NFκB were also lowered. Based on these findings, the researchers concluded that epalrestat inhibits AKR1B1, thereby suppressing the activation of the PKC/NFκB inflammatory pathway. This contributes to a decrease in inflammatory cytokine levels, renal tissue injuries, and systemic inflammatory response in the SA-AKI rat model. Therefore, AKR1B1 could be a potential target for the treatment of sepsis-associated acute kidney injuries.

The English expression of the manuscript can basically enable readers to understand the author's intention, but the writing level needs to be strengthened.

Experimental design

1.The objective of this research was to reduce the levels of inflammatory cytokines in serum and kidney tissues in a severe sepsis rat model. By identifying and targeting AKR1B1, the study proposed a novel therapeutic target for the prevention and treatment of sepsis and SA-AKI. Epalrestat, a commercially available Aldose Reductase Inhibitor (ARI), was identified as a promising candidate for treating SA-AKI. However, the effectiveness of Epalrestat in this context requires further clarification due to the complex nature of sepsis and SA-AKI pathogenesis and pathophysiological processes.
2.Thereafter, aldose reductase (AKR1B1) was screened as the study target. In the sepsis-associated acute kidney damage rat model, the animals were treated with the aldose reductase inhibitor (ARI), epalrestat. It is necessary to clarify and refine the screening methods of AKR1B1.
3.The application and discovery of proteomics in SA-AKI need to be discussed in detail in Introduction.
4.In the introduction, AKR1B1 is introduced, suggesting that AKR1B1 could be used as a potential therapeutic target for SA-AKI. Only later did the authors mention that they identified Aldose reductase (AKR1B1) as a potential therapeutic target for subsequent animal experiments by a label-free LC-MS/MS proteomics technique. The whole logical feeling is to first determine the research goal, and then to clarify this research goal through proteomics analysis. It is more logical to find this target through proteomics and then systematically study it.
5.Some methods need to be more detailed, such as primary and secondary antibodies in WB and Statistical Analysis.

Validity of the findings

1.The proteomic analysis identified a total of 5367 proteins in the kidney samples, out of which 4448 proteins were quantified. The differentially expressed proteins were compared between the septic and sham-operated groups. In both the moderate sepsis (A) and severe sepsis (B) rat AKI models, 17 proteins were found to be upregulated, while 9 proteins showed significant downregulation. The contents of the results need to be described in detail.
2.If P value≤0.05, any variation in the differential expression of >1.5 was employed as the criterion of change to indicate significant up-regulation, while values <1/1.5 were employed as the criterion of change to indicate significant down-regulation . The author should introduce a reference to confirm the filtered parameter of >1.5 or <1/1.5.
3.After performing a raw letter analysis and literature review analysis, AKR1B1 was selected as the candidate molecule. It showed a significant upregulation and its expression level was verified using a Western blot, which was consistent with the proteomic identification results (Figure 2). The authors should explain the detailed raw letter analysis and literature review analysis that contribute to the study of AKR1B1.
4.The results indicate that epalrestat inhibits the activation of the AKR1B1/PKC/NFκB pathway in the SA-AKI rat model, resulting in a reduction in AKR1B1, PKC-α, and NFκB p65 protein expression levels. This suggests that epalrestat has a suppressive effect on the inflammatory response and may be a potential therapeutic option for sepsis-associated acute kidney injuries. The authors need to explain why the PKC/NFκB signaling pathway was chosen for the study.
5.It is important to acknowledge that the signaling pathways involving AKR1B1, PKC, and NFκB are complex, and this study does not fully elucidate the detailed mechanisms underlying their interactions in SA-AKI. Further in-depth investigations and discussion are required to comprehensively understand and validate these findings.
6.Figure legends needs to be described in more detail.
7.In summary, while the proteomics-based screening of AKR1B1 in SA-AKI is a promising approach, further studies are needed to overcome the aforementioned limitations and validate AKR1B1 as a viable therapeutic target for SA-AKI. A comprehensive understanding of the biology and functional role of AKR1B1 in SA-AKI pathogenesis is necessary before considering its adoption as a therapeutic intervention in clinical practice. Validating the findings in a larger cohort of patients with SA-AKI and assessing the specificity, sensitivity, and predictive value of AKR1B1 as a biomarker is crucial before considering its use in clinical practice. In addition, further investigations, such as in vitro and in vivo experiments, are necessary to determine the precise role of AKR1B1 in SA-AKI pathogenesis and assess the effects of targeting AKR1B1 on disease outcomes.

Reviewer 3 ·

Basic reporting

1.Since sepsis-associated acute kidney injury (SA-AKI) has an extremely high morbidity and mortality rate worldwide, thus the aim of this study focuses on the development of new strategies for the treatment of SA-AKI from the point of view of revealing therapeutic targets. In this study, we used the SA-KAI rat model to identify the protein expression profiles of renal tissues using proteomics technology, and then screened for the potential therapeutic target of SA-AKI, AKR1B1, and followed up with tissue staining experiments and molecular assays to clarify the regulation of renal tissue injury and blood cytokines, thereby revealing the value of this gene in the clinical treatment of SA-AKI. In summary, this study is rigorous and reproducible, but it is recommended to further improve the writing level before publication.
2.The references should be updated, especially the old ones should be replaced with the new ones as far as possible.
3.The structure of the article is relatively complete, the chart is clear, and the original data can be corresponding.

Experimental design

4.It is suggested to add a link between sepsis and SA-AKI pathogenesis in the Introduction section, or to add a pathogenic mechanism, as describing sepsis at the beginning is slightly abrupt.
5.Because the goal of this paper is the development of novel therapeutic strategies for SA-AKI, it is therefore recommended that current therapeutic strategies for the palliation of SA-AKI be selected for in-depth description in the Introduction section.
6.Please show the type of rats used in this study in Materials and Methods, and please detail the surgical procedure, such as what part of the cecum is punctured, to improve the reproducibility of the experiment.
7.It has been described in line 273-275 that this study mined 17 up-regulated proteins and 9 down-regulated proteins by proteomic analysis, please elaborate on how these proteins will be useful for subsequent studies to better the transition.
8.The relevant description of Figure 2 does not indicate why AKR1B1 was identified as a candidate molecule for subsequent studies, please provide a reasonable explanation.

Validity of the findings

9.Proteomics provides a basis for the laws of life activity, but this paper does not add to the complete literature on proteins, and should add a fuller account of proteins used in the treatment of existing renal injury diseases, rather than skimming over them in the discussion section.
10.Please clarify in the text the special characteristics of AKR1B1 compared to other proteins in this paper, what exactly is the significance of its in-depth study, and please reflect the mining process of AKR1B1 in the Discussion section.
11.The conclusions of this paper lack a summary statement and transition to the introduction of later studies, for example, Figure 4 only explains the overexpression of inflammatory cytokines in the CLP group, but does not put forward a reasonable hypothesis, which is recommended to be added to enhance the readability of this paper.

Additional comments

12.The pathological results of Figure 5A need to be added with necessary scales.

---

## Round 0.2 · accepted · Accept

The reviewer's and my concerns were well addressed, and I think this revised version can be considered for publication in this journal, except for the following minor issues that need to be corrected during the Proof stage:

1. Page 1 and Page 3: In the Abstract section, the term "ARI preventative and" is unclear and should be removed for better clarity.

2. Fig. 1C is not mentioned in the main text. Please incorporate Fig. 1C in line 265.

3. The titles of Table 1 and Table 2 legends do not align with the content of the main text. Thus, the title of Table 1 (page 25) should be revised to [Gene-specific primers for real-time RT-PCR], and the title of Table 2 (page 27) should be revised to [In both the moderate sepsis (A) and severe sepsis (B) rat S-AKI models, 17 proteins were upregulated and seven were downregulated].

4. Page 1 and Page 3: "Blood urea ntrogen (BUN)" should be corrected to "Blood urea nitrogen (BUN)".

5. Line 90: "AKR1BI" should be corrected to "AKR1B1".

6. Line 134 and 135: "administered medicine before moulding" and "administered medicine after moulding" should be corrected to "administered medicine before modeling" and "administered medicine after modeling" respectively.

Reviewer 2 ·

Basic reporting

The author's revision addressed my concerns and suggested that the manuscript be accepted.

Experimental design

no comment

Validity of the findings

no comment

Reviewer 3 ·

Basic reporting

no comment

Experimental design

no comment

Validity of the findings

no comment

Additional comments

The Authors’ revisions may be accepted.